# Transcriptome, Ectopic Expression and Genetic Population Analysis Identify Candidate Genes for Fiber Quality Improvement in Cotton

**DOI:** 10.3390/ijms24098293

**Published:** 2023-05-05

**Authors:** Zhengwen Liu, Zhengwen Sun, Huifeng Ke, Bin Chen, Qishen Gu, Man Zhang, Nan Wu, Liting Chen, Yanbin Li, Chengsheng Meng, Guoning Wang, Liqiang Wu, Guiyin Zhang, Zhiying Ma, Yan Zhang, Xingfen Wang

**Affiliations:** State Key Laboratory of North China Crop Improvement and Regulation, North China Key Laboratory for Crop Germplasm Resources of Education Ministry, Hebei Agricultural University, Baoding 071001, China; liu889@126.com (Z.L.); nxszhw@hebau.edu.cn (Z.S.); kehuifeng123@126.com (H.K.); chbhebau@163.com (B.C.); guqishen0918@163.com (Q.G.); zhangman920601@163.com (M.Z.); wn2013@126.com (N.W.); mhyzh@hebau.edu.cn (G.Z.); mzhy@hebau.edu.cn (Z.M.)

**Keywords:** cotton, fiber development, transcriptome, overexpression, SNP markers, BC_3_F_5_ population

## Abstract

Comparative transcriptome analysis of fiber tissues between *Gossypium barbadense* and *Gossypium hirsutum* could reveal the molecular mechanisms underlying high-quality fiber formation and identify candidate genes for fiber quality improvement. In this study, 759 genes were found to be strongly upregulated at the elongation stage in *G. barbadense*, which showed four distinct expression patterns (I–IV). Among them, the 346 genes of group IV stood out in terms of the potential to promote fiber elongation, in which we finally identified 42 elongation-related candidate genes by comparative transcriptome analysis between *G. barbadense* and *G. hirsutum*. Subsequently, we overexpressed *GbAAR3* and *GbTWS1*, two of the 42 candidate genes, in *Arabidopsis* plants and validated their roles in promoting cell elongation. At the secondary cell wall (SCW) biosynthesis stage, 2275 genes were upregulated and exhibited five different expression profiles (I–V) in *G. barbadense*. We highlighted the critical roles of the 647 genes of group IV in SCW biosynthesis and further picked out 48 SCW biosynthesis-related candidate genes by comparative transcriptome analysis. SNP molecular markers were then successfully developed to distinguish the SCW biosynthesis-related candidate genes from their *G. hirsutum* orthologs, and the genotyping and phenotyping of a BC_3_F_5_ population proved their potential in improving fiber strength and micronaire. Our results contribute to the better understanding of the fiber quality differences between *G. barbadense* and *G. hirsutum* and provide novel alternative genes for fiber quality improvement.

## 1. Introduction

Cotton is an important economic crop which provides natural fiber materials for the textile industry [1]. *Gossypium hirsutum*, the most widely cultivated cotton species, accounts for more than 90% of the global cotton production because of its high yield and wide applicability. *Gossypium barbadense*, the other cultivated allotetraploid cotton species, is characterized by its superior fiber quality [2,3]. Breeders have been trying hard to develop cotton varieties with high yield, wide adaptability and superior fiber properties by crossing *G. barbadense* with *G. hirsutum*, but it is laborious and inefficient owing to the complex trait segregation of hybrid progenies. A more economic and effective strategy is molecular breeding, including molecular marker-assisted selection, transgene and gene editing, which requires a wide identification of fiber development-related genes and accurate verification of their functions.

Comparative transcriptome analysis has great potential in revealing molecular mechanisms underlying cotton fiber development and identifying candidate genes for fiber quality improvement. Various technologies have been developed to quantify the transcriptome, including microarray [4] and tag-based [5] methods. However, microarray methods have several limitations, including reliance upon the existing gene sequences, high background levels and a limited dynamic range of detection. Tag-based methods are also defective because a significant part of short tags cannot be uniquely mapped to the reference genome. By comparison, RNA-seq (massively parallel cDNA sequencing) has partly eliminated the challenges and provided a far more precise measurement of gene expression levels [6]. The RNA-seq of cotton fibers has been conducted for about ten years. Comparative transcriptome analysis based on RNA-seq has mainly been performed between superior and inferior upland cotton varieties (or lines) [7,8,9,10,11], between mutant (fuzzless, lintless or fuzzless–lintless) and wild-type varieties [12,13,14,15,16,17,18,19] and between chromosome segment substitution lines (a *G. hirsutum* background with *G. barbadense* introgression) and the recurrent parent [20,21,22]. These studies have contributed to dissecting cotton fiber development. *G. barbadense* and *G. hirsutum* clearly have different fiber properties, and thus a comparison of their fiber transcriptomes should also be an effective way to discover candidate genes for improving fiber quality. Nevertheless, as far as we can see, there are only a few studies that have conducted the comparison of the fiber transcriptomes of *G. barbadense* and *G. hirsutum* based on RNA-seq. The fiber transcriptomes of *G. barbadense* and *G. hirsutum* at 10 and 22 DPA were sequenced and compared in 2012, and high-quality reads from four libraries were assembled into 46,072 unigenes, functioning as the reference sequences [23]. The transcripts generated from the *G. raimondii* genome and the ESTs from *G. arboreum* were integrated to produce a mapping reference, and then the fiber transcriptomes of 10, 15, 18, 21 and 28 DPA were compared within and across genotypes (namely *G. barbadense* and *G. hirsutum*) in 2015 [24]. Recently, a series of released allotetraploid cotton whole-genome sequences has made it possible to apply a high-quality mapping reference to comparing the *G. barbadense* and *G. hirsutum* fiber transcriptomes. Moreover, the two studies mentioned above put the emphasis on hundreds of differentially expressed genes (DEGs), aiming to reveal the molecular mechanisms underlying cotton fiber development. Obviously, it still has a wide prospect to identify candidate genes by a comprehensive and ingenious comparison of the *G. barbadense* and *G. hirsutum* fiber transcriptomes.

Cotton fiber is a highly elongated and thickened unicellular trichome on the ovule surface whose development consists of four distinct but overlapping stages: fiber initiation (lint initiation at ~0 DPA; fuzz initiation at ~4 DPA), elongation (3–20 DPA), SCW biosynthesis (16–40 DPA) and maturation (40–50 DPA) [25,26]. Basically, fiber length is generated through primary cell wall (PCW) biosynthesis in the elongation stage, while the SCW biosynthesis stage forms a solid foundation for fiber strength and micronaire. *G. barbadense*, compared with *G. hirsutum*, possesses a higher rate of lint elongation and a prolonged elongation period, which gives rise to a greater length of mature fibers. However, secondary wall thickening in the two cotton species begins at a quite similar time, namely 16–20 DPA in *G. barbadense* and 16–19 DPA in *G. hirsutum* [27,28]. Consequently, we sequenced the fiber transcriptomes of initiation (represented by 0 DPA), elongation (represented by 5, 10 and 15 DPA) and SCW biosynthesis (represented by 20, 25 and 30 DPA) of *G. barbadense* and *G. hirsutum* in this study. Gene expression profiling was performed based on the *G. hirsutum* TM-1 genome [29] and subsequently comparative transcriptome analysis was conducted within and across genotypes. In addition to revealing the genetic basis for high-quality cotton fiber formation, more importantly, we further identified elongation-related and SCW biosynthesis-related candidate genes for fiber quality improvement. Transgenic *Arabidopsis* plants, SNP molecular markers and a BC_3_F_5_ population were then applied to validate candidate genes involving cotton fiber development. The identified candidate genes and developed SNP markers can facilitate future molecular breeding programs for improving cotton fiber quality.

## 2. Results

### 2.1. Overview of Fiber Transcriptomes

In the present study, to identify cotton fiber development-related genes, *G. barbadense* acc. Pima90-53 and Hai7124 with superior fiber properties and *G. hirsutum* acc. HY405 and ND13 with good fiber properties but inferior to *G. barbadense* were selected for comparative transcriptome analysis. Fiber quality analysis showed that the two *G. barbadense* accessions compared with the two *G. hirsutum* accessions possessed significantly higher fiber length (i.e., fiber upper-half mean length) and fiber strength and a lower fiber micronaire. Besides that, fiber elongation and fiber uniformity showed no significant differences (Figure 1a–d; Appendix A). In total, 28 cDNA libraries from seven timepoints were constructed to perform RNA-seq, including 0, 5, 10, 15, 20, 25 and 30 DPA (Figure 1e). Based on the time course of fiber development [25,26,27,28], 0 DPA (ovule samples) was designated as lint initiation. Moreover, lint elongation was referred to the fiber samples of 5, 10 and 15 DPA, while SCW biosynthesis was characterized using the fiber samples of 20, 25 and 30 DPA. A total of 420, 430, 437 and 414 million clean reads were generated in Pima90-53, Hai7124, HY405 and ND13, respectively. Although multi-mapped reads account for approximately 11% of the clean reads due to the high homology between the At and Dt subgenomes, about 80% of the clean reads were uniquely mapped to the upland cotton TM-1 genome (Appendix A). Specifically, the percentages varied from 75.74% to 79.63% in Pima90-53, from 74.50% to 80.03% in Hai7124, from 78.70% to 83.03% in HY405 and from 79.05% to 82.94% in ND13. According to the uniquely mapped reads, gene expression analysis was carried out for each sample, and then we performed principal component analysis (PCA) based on gene expression levels to analyze the correlations between the samples. Analyzed by PC1 and PC2, the samples were classified by development processes regardless of accessions. More specially, the samples of 20, 25 and 30 DPA clustered together, corresponding to the continuous and stable deposition of cellulose during SCW biosynthesis (Figure 1f). Analyzed by PC1 and PC3, the samples of *G. barbadense* were clearly separated from those of *G. hirsutum*, suggesting great potential to mine candidate genes for cotton fiber quality improvement by comparative transcriptome analysis.

### 2.2. Preferentially Expressed Genes during Fiber Elongation in G. barbadense

To identify the genes related to cotton fiber elongation and SCW biosynthesis in *G. barbadense*, differential expression analysis was conducted, which found a total of 17,473 and 16,538 genes (in Pima90-53 and Hai7124, respectively) to be significantly upregulated or downregulated at 5, 10, 15, 20, 25 and 30 DPA as compared with 0 DPA. Cluster analysis divided these differentially expressed genes into six categories based on expression trends (Appendix A). By comparison with lint initiation (0 DPA), type I genes were upregulated during lint elongation, type III genes were upregulated during SCW biosynthesis and type II genes showed an upregulated expression at both of these stages. On the contrary, type IV and VI genes exhibited a downregulated expression during lint elongation and SCW biosynthesis, respectively, and type V genes were downregulated at the two stages. Furthermore, the common genes of Pima90-53 and Hai7124 in each category were characterized with GO enrichment analysis to identify the significantly enriched biological processes (Appendix A). As a result, GO terms including fatty acid metabolic process, cellular carbohydrate metabolic process, plant type secondary cell wall biogenesis and actin cytoskeleton organization were enriched in the upregulated genes, while the downregulated genes were mainly enriched in GO terms involving cell proliferation and differentiation, such as chromatin organization, DNA metabolic process and meristem development. Obviously, the candidate genes promoting fiber elongation and/or SCW biosynthesis could be easily available from the upregulated genes rather than from the downregulated genes that may affect ovule development and fiber initiation.

Differential expression analysis identified 965 and 906 genes, whose expression was upregulated at 5, 10 and 15 DPA in comparison with 0 DPA, in *G. barbadense* acc. Pima90-53 and Hai7124, respectively (Figure 2a). The common 759 genes were subjected to cluster analysis and subsequently divided into four groups (Figure 2b; Appendix A). Although upregulated during fiber elongation, group I containing 55 genes exhibited higher expression levels during SCW biosynthesis, indicating that these genes may play more important roles in SCW thickening. Correspondingly, these genes were enriched in the carbohydrate transport and cellulose biosynthesis processes, based on GO enrichment analysis (Figure 2c). The 86 genes of group II were highly expressed during both fiber elongation and SCW biosynthesis, suggesting a solid foundation involved in cell growth. Roughly, the 272 genes of group III and the 346 genes of group IV possessed similar expression trends, which were preferentially expressed during fiber elongation. Group III genes were mainly enriched in actin cytoskeleton organization, carbohydrate catabolic process, response to salt stress and transmembrane transport, indicating a potential role in cell expansion. Unlike group III genes, members in group IV were upregulated only during fiber elongation, not SCW biosynthesis. GO enrichment analysis showed that a lot of fiber elongation-related processes were enriched in group IV genes, including the S-adenosylmethionine metabolic process, very long-chain fatty acid metabolic process and water transport. Furthermore, group IV genes were divided into three subgroups (IV-I, -II and -III), based on the time at which their expression reached the peak (Figure 2b). The IV-I genes exhibited the highest expression at 5 and 10 DPA and were mainly enriched in the S-adenosylmethionine metabolic process, implying that these genes may create initial conditions for cotton fiber elongation. The expression of the IV-II genes peaked at 10 DPA, and the IV-III genes showed the highest expression at 10 and 15 DPA. Hence, these two subgroups might have contributed to the rapid elongation of fiber cells, which was also supported by the significantly enriched GO terms (Figure 2d).

### 2.3. Identification and Functional Analysis of Fiber Elongation-Related Genes

Considering the deep involvement in cotton fiber elongation, the 346 genes of group IV were further employed and then refined by comparing the *G. barbadense* and *G. hirsutum* fiber transcriptomes. As a result, we identified 42 genes whose expression was upregulated in *G. barbadense* as compared with *G. hirsutum* during cotton fiber elongation (Figure 2e; Appendix A). These genes consisted of 10 IV-I genes, 18 IV-II genes and 14 IV-III genes (Table 1). Among them, the *G. hirsutum* orthologs *Gbar_A11G021290* (3-hydroxyacyl-CoA dehydratase) [30], *Gbar_D04G001950* (cytochrome P450; GenBank: AJ606074), *Gbar_D05G037750* ((+)-δ-cadinene synthase) [31], *Gbar_D11G024050* (glutathione S-transferase; GenBank: AF159229) and *Gbar_D12G003220* (SAUR-like auxin-responsive protein; GenBank: KM065453) were cloned in previous studies, as was *Gbar_A07G001880* (pectinesterase) [32]. In addition, several genes were found in cotton fiber development-related gene families, such as *Gbar_A03G019080* (nonspecific lipid-transfer protein) [33], *Gbar_A07G007890* (α-expansin) [34] and *Gbar_D08G001620* (dirigent protein) [35].

To further validate the potential roles of the 42 candidate genes in cell elongation, *Gbar_A05G037170* and *Gbar_A05G020690* driven by a CaMV35S promoter were transferred independently into *A. thaliana* plants, and then their roles were determined by observing the changes of leaf trichomes and dark-grown hypocotyl cells. The two genes were selected by considering their potential as novel genes related to cotton fiber development. *Gbar_A05G037170* (termed *GbAAR3*) is highly homologous to the *AtAAR3* identified in a screen for mutants resistant to an anti-auxin [36]. Encoding a DCN1-like protein, *GbAAR3* may regulate cullin neddylation and thus participate in auxin signaling by the SCF^TIR1/AFB^ pathway [37,38]. As expected, transient expression analysis of *GbAAR3* fused to green fluorescent protein (GFP) confirmed its nuclear localization (Figure 3a). Two transgenic T_3_ lines OE2 and OE5 were developed, in which the stable expression of *GbAAR3* was confirmed by qRT-PCR (Figure 3b). *Arabidopsis* leaf trichomes were subsequently measured because they partly share common regulatory mechanisms with cotton fiber cells [39,40,41,42]. As a result, the transgenic OE2 and OE5 plants, compared with WT plants, showed significantly longer trichomes (Figure 3c,d). Moreover, dark-grown hypocotyls were employed because their growth results from cell elongation rather than division [43,44]. Here, five-day-old dark-grown OE2, OE5 and WT seedlings were measured, and then we discovered that the transgenic seedlings had significantly longer hypocotyls (Figure 3e,f). Correspondingly, longer hypocotyl epidermal cells were observed in the transgenic seedlings in a microscopic inspection (Figure 3g). Similarly, *Gbar_A05G020690* (named *GbTWS1*) is homologous to *AtTWS1* which encodes a novel small protein and is speculated to affect the functionality of the endoplasmic reticulum [45]. To determine the involvement of *GbTWS1* in cell elongation, *Arabidopsis GbTWS1*-overexpressed T_3_ lines OE5 and OE7 were generated, and stable expression was confirmed by qRT-PCR and Western blotting (Appendix A). Although the overexpression of *GbTWS1* in *Arabidopsis* plants promoted the elongation of leaf trichomes only slightly, the transgenic seedlings produced significantly longer dark-grown hypocotyl cells and thus longer hypocotyls in comparison with WT seedlings (Appendix A). These results indicate that *GbAAR3* and *GbTWS1* can promote cell elongation and might have contributed to cotton fiber development and that the 42 elongation-related candidate genes may have good prospects in fiber quality improvement.

### 2.4. Preferentially Expressed Genes during SCW Biosynthesis in G. barbadense

Differential expression analysis produced 2639 and 2568 genes that were upregulated at 20, 25 and 30 DPA as compared with 0 DPA in *G. barbadense* acc. Pima90-53 and Hai7124, respectively (Figure 4a). Subsequently, cluster analysis using gene expression data divided the common 2275 genes into five groups (Figure 4b; Appendix A). The 466 genes in group I were upregulated during SCW thickening but showed the highest expression during fiber elongation. A lot of GO terms were significantly enriched and mainly classified into five categories: cytoskeleton organization, ATP metabolism, cell wall development, gene expression regulation and transport (Appendix A). The 329 genes in group II exhibited high expression levels during both fiber elongation and SCW thickening. Interestingly, the group II genes were not enriched in biological processes associated with cell wall development, especially the biomacromolecule metabolic process (Appendix A). It was obvious that the biosynthesis of PCW and SCW featured distinct gene networks. The 686 genes in group III were preferentially expressed during SCW thickening and were slightly upregulated during fiber elongation. A number of cell wall development-related processes related to cellulose, hemicellulose and lignin were significantly enriched in the group III genes. Correspondingly, we observed a lot of significantly enriched GO terms involving transport, especially vesicle-mediated transport (Appendix A). The 647 genes in group IV and the 147 genes in group V had similar expression trends, i.e., only upregulated during SCW biosynthesis. The group IV genes were significantly enriched in the metabolic processes of cellulose, xylan, glucuronoxylan, lignin, pectin, arabinan and glycoprotein, which all belong to cell wall macromolecules (Appendix A). However, being different from the group III genes, the group VI members were hardly enriched in transport-related processes. The group V genes showed the highest expression at 30 DPA and thus were more likely to affect cotton fiber maturation (Appendix A). In addition, more GO terms involved in regulating gene expression were significantly enriched in the group I and II genes, presumably because the group III and IV genes were mainly responsible for continuous and stable SCW deposition. Obviously, all the five groups contributed to cotton fiber development based on the principle of coordination and unification (Figure 4c).

### 2.5. Identification and Functional Analysis of the SCW Biosynthesis-Related Genes

Given the critical roles of the group IV genes in SCW thickening (Figure 4d), the 647 genes were further used to identify candidate genes improving cotton fiber strength and micronaire. By means of comparative transcriptome analysis, 48 genes were filtered from the group IV genes which were upregulated in *G. barbadense* as compared with *G. hirsutum* during SCW biosynthesis (Figure 4e). The 48 candidate genes were subsequently divided into two categories based on their expression trends (Figure 4f). When the SCW biosynthesis stage was compared with the fiber initiation stage, the 25 genes of type I (Table 2 and Appendix A) were upregulated in both *G. barbadense* and *G. hirsutum*, but the 23 genes of type II (Table 3 and Appendix A) were only upregulated in *G. barbadense*. A number of well-known cell wall development-related genes were observed, including *Gbar_A10G023160* (UDP-Xyl synthase) [46], *Gbar_A11G034900* (laccase) [47], *Gbar_A13G023580* (fasciclin-like arabinogalactan protein) [48], *Gbar_D10G011020* (xylan glucuronosyltransferase) [49], *Gbar_D10G018450* (β-xylosidase) [50], *Gbar_A05G018470* (endo-β-1, 4-glucanase) [51], *Gbar_A07G004070* (cellulose synthase) [52] and *Gbar_D03G007510* (cellulose synthase) [52]. 

To further verify the involvement of the 48 candidate genes in fiber quality improvement, we performed the genotyping and phenotyping of a BC_3_F_5_ population developed from donor parent *G. barbadense* acc. Pima90-53 and recipient parent *G. hirsutum* acc. CCRI8, and then determined whether the introgression of the *G. barbadense* candidate genes into *G. hirsutum* could improve fiber properties. For the genotyping, SNP molecular markers of 39 candidate genes were successfully developed, which can distinguish the orthologous genes between *G. barbadense* and *G. hirsutum* accurately and easily (Appendix A). The primers of molecular markers were designed based on the SNPs between *G. barbadense* and *G. hirsutum* and the SNPs between the At and Dt subgenomes (Figure 5a; Appendix A). Subsequently, in the BC_3_F_5_ population, alleles were detected at each locus, showing three types of genotypes: *Gb*/*Gb* homozygote, *Gh*/*Gh* homozygote and *Gb*/*Gh* heterozygote (Figure 5b). The phenotype data collected from Luntai (E1) and Qingxian (E2) were thus compared based on different genotypes. Interestingly enough, the type I SCW biosynthesis-related candidate genes enhanced fiber strength and did not impair fiber length and micronaire while the type II candidate genes were much more likely to improve fiber micronaire (Figure 5c; Appendix A). *Gbar_A05G019310* belonging to type I shows similarity to *AtIDD1* that encodes a C2H2 zinc finger protein and mediates GA signaling. GA has been proposed to promote SCW deposition in cotton fiber cells [53]. Here, *Gbar_A05G019310* was preferentially expressed during SCW biosynthesis in *G. barbadense*, and its orthologous gene in *G. hirsutum* exhibited similar expression patterns but lower transcript levels, supported by the current transcriptome data and a supplementary RNA-seq project of *G. hirsutum* acc. CCRI8 (Figure 5d; Appendix A). As expected, it contributed to fiber strength improvement when Pima90-53 genome fragments containing *Gbar_A05G019310* were recombined into the CCRI8 genome. Similarly, another type I gene *Gbar_A05G025000*, encoding a hypothetical transmembrane protein with unknown functions, also enhanced fiber strength (Figure 5e). Comparative phenotype analysis also identified three type II genes *Gbar_A06G015550* (Figure 5f), *Gbar_A13G023450* (Figure 5g) and *Gbar_D05G026930* (Figure 5h) that improved fiber micronaire significantly. *Gbar_A06G015550* encodes the aldehyde dehydrogenase associated with cellular responses to oxidative stress [54]. Encoding the putative epoxide hydrolase, *Gbar_A13G023450* is potentially involved in transforming epoxide-containing fatty acids and thereby cutin biosynthesis [55]. *Gbar_D05G026930* encodes an uclacyanin-like blue copper-binding protein that has been implicated in lignin biosynthesis [56,57]. These kinds of genes are generally considered to involve stress responses, and thus the results provide a novel alternative way to improve cotton fiber quality. 

## 3. Discussion

In the present study, we identified 42 and 48 candidate genes for cotton quality improvement during fiber elongation and SCW biosynthesis, respectively. Among the 42 elongation-related candidate genes, some genes, e.g., α-expansin and pectinesterase, have been widely proposed to affect cotton fiber development (Table 1). Therefore, the identified candidate genes have paved alternative ways for fiber length improvement. *Gbar_A13G012640* from IV-I, encoding a DUF538 domain-containing protein, is homologous to *AtSVB* (*AT1G56580*), mutants of which exhibit smaller trichomes. *Arabidopsis* leaf trichomes share similar developmental mechanisms with fiber cells of cotton [39,40,41,42], and thus *Gbar_A13G012640* might be responsible for cotton fiber development. ROS (reactive oxygen species) signaling is well-known to be crucial for cell elongation [58]. Strikingly, two highly homologous genes *Gbar_A11G014750* and *Gbar_D11G015570* encode a cysteine-rich STOMAGEN peptide, showing great potential in ROS perception [59]. The two genes were predominantly expressed during early stages of fiber elongation (Figure 2b), and thus it is conceivable that they might contribute to initiating the elongation process. ROS, at high concentrations, can lead to cell wall stiffening and thus suppress extension [60]. Not surprisingly, two IV-II candidate genes *Gbar_D11G024050* (glutathione S-transferase) [61] and *Gbar_D12G026630* (heat shock protein) [62], showing the highest expression at 10 DPA, may play a significant role in ROS homeostasis and help to maintain the rapid elongation of fiber cells. In addition, *Gbar_A07G023310* from IV-III, encoding formate dehydrogenase, is also involved in scavenging ROS, as it produces NADH [63]. Remarkably, several genes participating in the metabolism of S-adenosylmethionine stood out, including *Gbar_A07G013810* (S-adenosylmethionine synthase), *Gbar_A09G015680* (S-adenosylmethionine synthase), *Gbar_D10G014320* (methylenetetrahydrofolate reductase) and *Gbar_A06G013440* (cystathionine β-synthase) [64]. S-adenosylmethionine not only provides the methyl group for the majority of methylation reactions, but also enters the ethylene and polyamine biosynthetic pathways. Obviously, it is worth trying to dissect the roles of the S-adenosylmethionine metabolism-related genes in fiber development. Besides that, the fact that *GbAAR3* (from IV-III) promoted cell elongation in *Arabidopsis* plants raises our interest on cullin neddylation as well as SCF^TIR1/AFB^-mediated auxin signaling in cotton fiber elongation [37,38] reinforced by a SAUR-like auxin-responsive protein (*Gbar_D12G003220* from IV-III). 

Among the 48 SCW biosynthesis-related candidate genes, a number of well-known cell wall development-related genes were observed and presented. Here, we discuss the functions of other genes which provide novel potential ways for fiber quality improvement. Unlike vascular cells, cotton fibers contain only a small amount of lignin whose role in fiber development has been neglected. Recent studies, however, have shown that lignin-like phenolics can affect fiber quality substantially in a negative pattern during elongation and in a positive pattern during SCW thickening [65,66]. Type I candidate gene *Gbar_A11G034900* encodes laccase that is responsible for the final stage of lignin polymerization [47]. Having higher expression levels than its *G. hirsutum* ortholog during SCW thickening (Figure 4f), *Gbar_A11G034900* may contribute to forming stronger and thinner fiber cells. Uclacyanin proteins have been considered to regulate lignin biosynthesis [56,57], but the mechanisms remain unclear. Hence, the fiber fineness improvement caused by the introgression of type II candidate gene *Gbar_D05G026930* encoding an uclacyanin-like blue copper-binding protein could be due to an increase in lignin/lignin-like phenolics (Figure 5h). Recent advances have revealed that plant hormones play a pivotal role in regulating cotton fiber development, mainly focusing on fiber initiation and elongation. Here, type I gene *Gbar_A05G019310* shows high homology to *AtIDD1* [67,68] and thus is assumed to mediate GA signaling. The fact that *Gbar_A05G019310* can enhance fiber strength (Figure 5d) highlights the importance of the IDD (INDETERMINATE DOMAIN) family as well as GA signaling in SCW biosynthesis. Another type I gene *Gbar_A05G022250* encodes the EXORDIUM protein that has been proposed to participate in BR signaling [69]. BR signaling appears likely to change the expression of cellulose synthase genes and thus affects SCW deposition in cotton fiber [70]. Intriguingly, we identified two cellulose synthase genes *Gbar_A07G004070* and *Gbar_D03G007510*, suggesting a possible signaling module involved in BRs, EXORDIUM protein and cellulose synthase. The ABC transporter plays a central role in transporting auxin, ABA and cytokinin. *Gbar_A10G008340* from type II, encoding the ABCG-type transporter, is homologous to *AtABCG40* that exhibits ABA uptake activity [71]. Another type II gene *Gbar_A13G001740* encodes the auxin influx transporter that is responsible for importing auxin into cells [38]. Obviously, phytohormones play a vital role in SCW biosynthesis, and their crosstalk and transcriptional regulation may contribute to stronger and thinner fibers in *G. barbadense*. Hormonal regulation of transcription is involved in the ubiquitin/26S proteasome pathway, such as Aux/IAA degradation in auxin signaling [38] and DELLA degradation in GA signaling [68]. *Gbar_A05G004360* from type I encodes the putative F-box component of the SCF E3 ligase complex, while type II gene *Gbar_D06G019950* encodes RING-type E3 ligase. It is particularly interesting to investigate whether these two genes act as key regulators of hormone signaling in plants.

In this study, genetic experiments were conducted to further confirm the roles of the candidate genes in cotton fiber development. By means of homology analysis, we speculated that *GbAAR3*, a type IV-III elongation-related candidate gene, could regulate auxin signaling through the cullin neddylation-mediated SCF^TIR1/AFB^ pathway [36,37,38]. *GbTWS1* from IV-II, which is implicated in the functioning of the endoplasmic reticulum, might affect fatty acid biosynthesis and thus regulate the organization of the endomembrane system [45]. The fact that overexpression of *GbAAR3* (Figure 3) and *GbTWS1* (Appendix A) can promote cell elongation in *Arabidopsis* has increased the importance of candidate genes in fiber length enhancement of cotton. For the SCW biosynthesis-related candidate genes, we developed SNP molecular markers to distinguish the candidate genes from their *G. hirsutum* orthologs, and then the genotyping and phenotyping of a BC_3_F_5_ population were conducted to evaluate the influence on cotton fiber quality generated by the introgression of the candidate genes into *G. hirsutum*. Although type I candidate genes enhanced fiber strength as a whole, several members showed pleiotropic effects. For instance, *Gbar_D06G000230*, a β-tubulin gene, exhibited dual functions in improving fiber strength and fiber micronaire (Appendix A), presumably because cortical microtubules play a vital role in cellulose microfibril deposition [72]. GA signaling in general promotes both elongation and SCW development in cotton fiber cells [73]. Correspondingly, the proposed GA signaling-related gene *Gbar_A05G019310* enhanced not only fiber strength but also length (Appendix A). Similarly, the hypothetical transmembrane protein Gbar_A05G025000 also improved fiber length and strength (Appendix A). Furthermore, *Gbar_A05G019310* fell into FUqQtlc05_1b [74], and *Gbar_A05G025000* fell around QTLClust_LEN_5_3 [75] when analyzing the QTLs reported for fiber length. In this light, we can infer that some SCW biosynthesis-related genes also contribute to fiber elongation, especially because a considerable portion of SCW thickening occurs before fiber elongation ceases in both *G. barbadense* [28] and *G. hirsutum* [27]. Type II gene *Gbar_A06G015550* encoding aldehyde dehydrogenase can oxidize a wide range of reactive and toxic aldehydes and meanwhile produce NAD(P)H, a highly effective antioxidant [54]. Hence, BC_3_F_5_ lines with the introgression of *Gbar_A06G015550* may enhance the capacity for the management of ROS and thus showed the enhancement of fiber length besides the abovementioned fiber micronaire improvement (Appendix A). Interestingly enough, another type II gene *Gbar_A13G023450* seemed to simultaneously improve fiber length, strength and micronaire (Appendix A). *Gbar_A13G023450* encodes epoxide hydrolase and is likely responsible for cutin biosynthesis [55]. Cutin is the main constituent of the cuticle, which is the outermost layer of cotton fibers and provides a physical barrier, but to date, there is little evidence that cutin biosynthesis affects cotton fiber development [76]. More genetic experiments by overexpressing and knocking out *Gbar_A13G023450* need to be performed to further confirm its remarkable functions. Taken together, the BC_3_F_5_ population analysis proved the importance of the candidate genes in fiber quality improvement and highlighted several pleiotropic genes. Nevertheless, it is worth noting that some candidate genes failed to be characterized because of no or little corresponding lines and that the influence on cotton fiber quality may be caused by candidate genes and their closely linked genes. A larger population will be applied to overcome this problem, and we will investigate the functions of the candidate genes by means of CRISPR/Cas9-mediated gene editing in future efforts.

## 4. Materials and Methods

### 4.1. Plant Materials and Growth Conditions

The cotton plant materials used in this study included *G. barbadense* acc. Pima90-53 [77] and Hai7124 [3] and *G. hirsutum* acc. HY405 [35], ND13 [35] and CCRI8. Pima90-53 (accession number: M210080; introduced from the USA), Hai7124 (M210054; Jiangsu, China) and CCRI8 (M110553; Henan, China) were collected and preserved with the appropriate permissions by the National Medium-Term Gene Bank of Cotton in China. HY405 (G100937; Hebei, China) and ND13 (G100728; Hebei, China) were collected and preserved by Hebei Agricultural University. All the necessary permissions for planting and investigating these cultivars were obtained from Hebei Agricultural University and the National Medium-Term Gene Bank of Cotton in China. The collection and research of these cultivars complied with the Convention on International Trade in Endangered Species of Wild Fauna and Flora. A BC_3_F_5_ population consisting of 167 lines was constructed and used in this study. Firstly, *G. barbadense* acc. Pima90-53 as the male parent was crossed with *G. hirsutum* acc. CCRI8 to generate an F_1_ hybrid in Baoding, Hebei Province, China, during summer 2009. The hybrid plants as the female parent were then continuously backcrossed with CCRI8 for three generations to produce BC_3_F_1_ plants in Baoding from 2010 to 2012. Eventually, to raise BC_3_F_5_ plants, the BC_3_F_1_ plants were self-pollinated for four generations in summer 2013 (Baoding), winter 2013 (Sanya, Hainan, China), summer 2014 (Baoding) and summer 2015 (Baoding). Molecular marker-assisted selection was performed at the BC_3_F_1_–BC_3_F_3_ generations to ensure the introgression of *G. barbadense* into *G. hirsutum*. 

For transcriptome sequencing, Pima90-53, Hai7124, HY405 and ND13 were grown at the cotton breeding center (38°45′ N, 115°29′ E) in Baoding, with a warm temperate continental monsoon climate, from late April to late October in 2014. Each plot consisted of six rows of 7 m in length containing 20–22 plants per row, with 30–35 cm between the plants within each row and 80 cm between the rows. The 167 BC_3_F_5_ lines and their parents were planted in a randomized complete block design including three replications in Qingxian, Hebei Province (38°65′ N, 116°91′ E), with a temperate semi-humid continental monsoon climate, and in Luntai, Xinjiang Uygur Autonomous Region (41°46′ N, 84°14′ E), with a warm temperate continental arid climate, in 2016. In Qingxian, each plot contained one row of 7 m in length, with 20–22 plants per row, 30–35 cm between the plants within each row and 76 cm between the rows, whereas in Luntai, each plot contained one row of 7 m in length containing 60–66 plants per row, with about 10 cm between the plants within each row and 76 cm between the rows. The population materials were sown in mid- to late April and were harvested in mid- to late October. All field management including watering, weed management and fertilization was performed according to the local production practices throughout the growth period. *A. thaliana* (Columbia ecotype) wild-type and transgenic plants were grown in pots containing sterile vermiculite in a greenhouse (22 °C, 16 h photoperiod, 70% relative humidity) at Hebei Agricultural University, with Hoagland’s nutrient solution added weekly.

### 4.2. Transcriptome Sequencing

Flowers of *G. barbadense* acc. Pima90-53 and Hai7124 and *G. hirsutum* acc. HY405 and ND13 were tagged at the flowering stage (from mid-July to late August). For each accession, ovule samples of 0 DPA and fiber samples of 5, 10, 15, 20, 25 and 30 DPA were collected independently from cotton bolls, frozen immediately in liquid nitrogen and ground mechanically to a fine powder. For each timepoint, samples from multiple cotton plants were pooled to minimize variations. Total RNA was then isolated using an RNAprep pure Plant Kit (TIANGEN, Beijing, China). Finally, 28 cDNA libraries were constructed using a NEBNext Ultra RNA Library Prep Kit for Illumina (NEB, Ipswich, MA, USA) at the Novogene Bioinformatics Institute, Beijing, China. Briefly, mRNA was first purified and fragmented. Secondly, cDNA was synthesized using a random hexamer primer, and the sequencing adaptor was ligated. Thirdly, fragments containing an insert of 150–200 bp were selected, and PCR amplification was performed. 

An Illumina Hiseq 2500 platform was then used for the sequencing, and 125 bp paired-end reads were generated. To generate clean reads, raw data were first processed by removing the adapter- or polyN-containing reads and the reads with low quality. The clean reads were then aligned to the *G. hirsutum* TM-1 genome [29] using TopHat2 (version 2.0.12; Toronto, Ontario) [78] with the threshold of two mismatches, and the mapped reads were assembled by Cufflinks (version 2.1.1; Dallas, Texas, United States) [79]. HTSeq (version 0.6.1; Wilmington, Delaware, United States) [80] was used to count the reads mapped to each gene, and subsequently RPKM (reads per kilobase of exon model per million mapped reads) was applied to estimate the expression levels. After the read counts were adjusted using the edgeR package (version 3.0.8; Wilmington, Delaware, United States) [81], differential expression analysis was performed using the DEGSeq package (version 1.12.0; Wilmington, Delaware, United States) [82], with the criteria of |log_2_(fold change)| > 1 and Q-value < 0.005. GO enrichment analysis was implemented using KOBAS v3.0 [83]. The *p*-value generated from the hypergeometric test was adjusted via the Benjamini–Hochberg method, generating the Q-value. GO terms with the Q-value < 0.05 were considered significantly enriched. 

### 4.3. Functional Analysis of the Elongation-Related Candidate Genes

The functional identification of cotton fiber elongation-related candidate genes in *A. thaliana* plants was conducted using the methods described previously [35]. To be specific, the coding sequence of candidate genes was cloned from fibers of *G. barbadense* acc. Pima90-53 and inserted into the Gateway pDONR207 vector to generate an entry clone. The gene clone was then recombined into the Gateway pGWB414 vector to construct a constitutive overexpression (OE) system under the control of a CaMV35S promoter. The OE system was transferred with an *Agrobacterium*-mediated method into *A. thaliana* Columbia ecotype plants, and subsequently the transgenic plants were confirmed by kanamycin-resistant selection and PCR detection. To observe the changes of trichomes, the fifth rosette leaves of four-week-old *A. thaliana* wild-type (WT) and OE plants (T_3_ homozygous lines) were harvested, decolorized by ethanol and photographed with an Olympus BX51 microscope (Tokyo, Japan). The longest branch of about 150 legible trichomes was measured using the ImageJ software [84] in each line. To compare dark-grown hypocotyls, the seeds of *A. thaliana* WT and OE plants were sterilized and then grown in vertical plates (1/2 MS medium, 0.9% agar and pH 5.8) under the conditions of 22 °C and continuous darkness. Five-day-old seedlings were photographed with a professional Epson V800 scanner (Nagano, Japan), and their hypocotyls were then measured using the ImageJ software. Stained with propidium iodide (PI), the seedlings used above were observed using an Olympus FV10i laser scanning microscope (Tokyo, Japan) to determine the epidermal cells of the hypocotyls. The GraphPad Prism software (San Diego, CA, USA) was employed to conduct a two-tailed *t*-test, and *p*-values < 0.05 were considered statistically significantly different. To determine the subcellular localization, the gene clone was recombined into the Gateway pEarleyGate103 vector to express the target protein with a C-terminal GFP fusion. The GFP-fused target protein and GFP as a control were transiently expressed in onion epidermal cells by means of a Bio-Rad PDS-1000/He system (Hercules, CA, USA). Incubated on an MS agar medium for 24 h in continuous darkness, the transformed cells were observed and photographed with an Olympus BX51 microscope (Tokyo, Japan).

### 4.4. Functional Identification of the SCW Biosynthesis-Related Candidate Genes

The BC_3_F_5_ population was employed to determine the involvement of the SCW biosynthesis-related candidate genes in fiber quality improvement. Twenty mature bolls of each accession (167 BC_3_F_5_ lines and their parents) were harvested from the middle fruiting branches of cotton plants, and the phenotyping of fiber quality traits including length, strength, micronaire, elongation and uniformity was performed using 15 g fiber samples on an Uster HVI 1000 system under environmental conditions of 20 °C and 65% relative humidity. For genotyping, fresh leaf tissue of each accession was used for genomic DNA isolation using a modified CTAB method [85]. To differentiate the SCW biosynthesis-related candidate genes from their *G. hirsutum* orthologs, a simple and cost-effective tri-primer AS-PCR [86] assay was employed based on in silico single-nucleotide polymorphism (SNP) identification. For each candidate gene locus, the 3’ end of primers P1 and P2 was designed for the SNP between *G. barbadense* and *G. hirsutum*, and the 3’ end of a third primer P3, common to both species, was designed for the SNP between the At and Dt subgenomes to avoid effects from highly homologous genes. The *G. barbadense*-specific primer pair (P1 and P3) and the *G. hirsutum*-specific primer pair (P2 and P3) were each used for population genotyping. The *Gb*/*Gb* homozygote showed PCR products only in P1–P3, the *Gh*/*Gh* homozygote showed PCR products only in P2 and P3 and the *Gb*/*Gh* heterozygote showed PCR products in both primer pairs. To determine the influence on fiber quality with the introgression of the candidate genes, the phenotype data of accessions with the *Gh*/*Gh* genotype were compared with those of accessions with the *Gb*/*Gb* or *Gb*/*Gh* genotype. The significance of difference was analyzed with a two-tailed *t*-test.

## 5. Conclusions

In the present study, we highlighted a batch of genes preferentially expressed during fiber elongation and SCW biosynthesis, and thus further revealed the genetic basis underlying high-quality fiber formation. By comparative transcriptome analysis between *G. barbadense* and *G. hirsutum*, we finally identified 42 elongation-related and 48 SCW biosynthesis-related candidate genes, whose potential roles were then confirmed by ectopic overexpression and SNP marker-based BC_3_F_5_ population analysis, respectively. These findings not only provide valuable information for the understanding of cotton fiber development, but also credibly contribute to cotton breeding for fiber quality improvement.

## Figures and Tables

**Figure 1 ijms-24-08293-f001:**
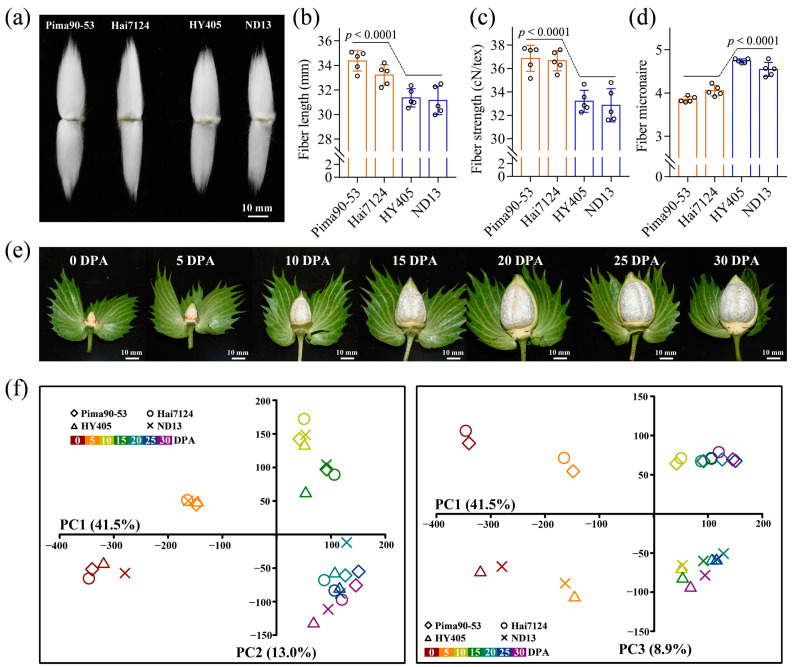
Transcriptome sequencing of the *G. barbadense* and *G. hirsutum* fibers. (**a**) Mature cotton fibers from *G. barbadense* and *G. hirsutum*. (**b**) Difference in fiber length between the two cotton groups. Phenotype data were collected from five years × location agroecological environments. The significance of difference was analyzed with a two-tailed *t*-test. (**c**) Difference in fiber strength between the two cotton groups. (**d**) Difference in fiber micronaire between the two cotton groups. (**e**) Cotton boll development in *G. barbadense*. (**f**) Principal component analysis of the sequencing samples based on gene expression.

**Figure 2 ijms-24-08293-f002:**
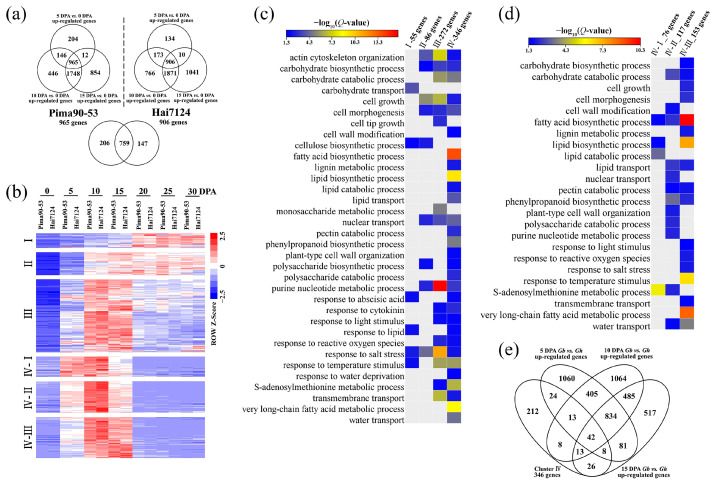
Preferentially expressed genes during cotton fiber elongation in *G. barbadense*. (**a**) Identification of the DEGs that were upregulated at 5, 10 and 15 DPA in comparison with 0 DPA. (**b**) Cluster analysis of the upregulated DEGs based on expression trends. The RPKM values were normalized with the z-score method. (**c**) GO enrichment analysis in different clusters; *p*-value generated from the hypergeometric test was adjusted via the Benjamini–Hochberg method, generating a *Q*-value; GO terms with a *Q*-value < 0.05 were considered significantly enriched. (**d**) GO enrichment analysis in the IV-I, IV-II and IV-III subgroups. (**e**) Forty-two genes were extracted from cluster IV, which were upregulated in *G. barbadense* as compared with *G. hirsutum* during cotton fiber elongation.

**Figure 3 ijms-24-08293-f003:**
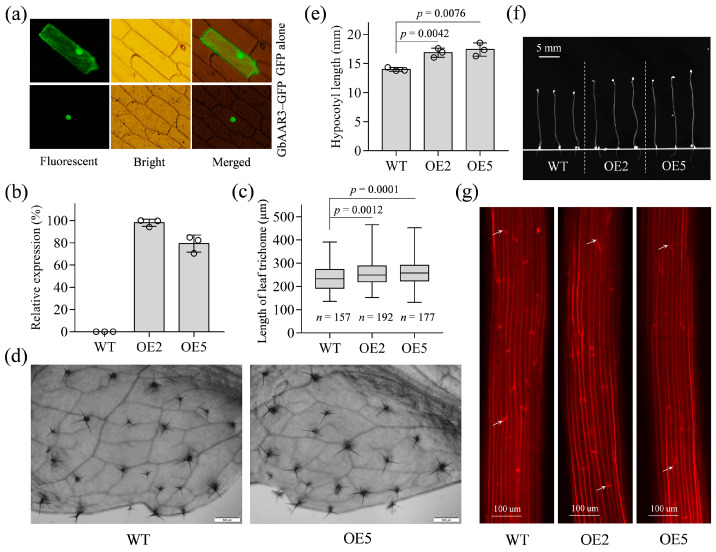
Functional analysis of *GbAAR3* in *Arabidopsis*. (**a**) Subcellular localization of the *GbAAR3*–GFP protein in onion epidermal cells. GFP alone was used as the control. (**b**) Expression of *GbAAR3* in *GbAAR3*-overexpressed *Arabidopsis* plants as detected by qRT-PCR. (**c**,**d**) *Arabidopsis* plants with overexpression of *GbAAR3* compared with the wild type showed significantly longer leaf trichomes. In the boxplots, the center line indicates the median, the box limits denote the upper and lower quartiles and the whiskers mark the range of data; *n* shows the number of measured trichomes. The significance of difference was analyzed with a two-tailed *t*-test. (**e**,**f**) *Arabidopsis* plants with overexpression of *GbAAR3* compared with the wild type showed significantly longer dark-grown hypocotyls. (**g**) Microscopic inspection of hypocotyl epidermal cells. The arrows mark the ends of representative epidermal cells.

**Figure 4 ijms-24-08293-f004:**
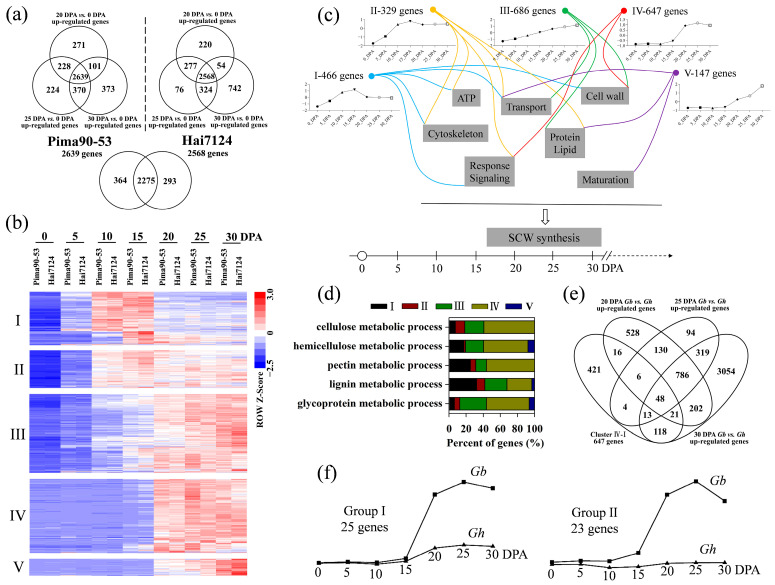
Preferentially expressed genes during SCW biosynthesis in *G. barbadense*. (**a**) Identification of the DEGs that were upregulated at 20, 25 and 30 DPA in comparison with 0 DPA. (**b**) Cluster analysis of the upregulated DEGs based on expression trends. The RPKM values were normalized with the z-score method. (**c**) Comprehensive diagram illustrating the involvement of different clusters in SCW biosynthesis. (**d**) Statistics on the genes related to cell wall macromolecule metabolism in different clusters. (**e**) The 48 genes extracted from cluster IV which were upregulated in *G. barbadense* as compared with *G. hirsutum* during SCW biosynthesis. (**f**) Forty-eight genes were divided into two groups based on the expression patterns of their *G. hirsutum* orthologs.

**Figure 5 ijms-24-08293-f005:**
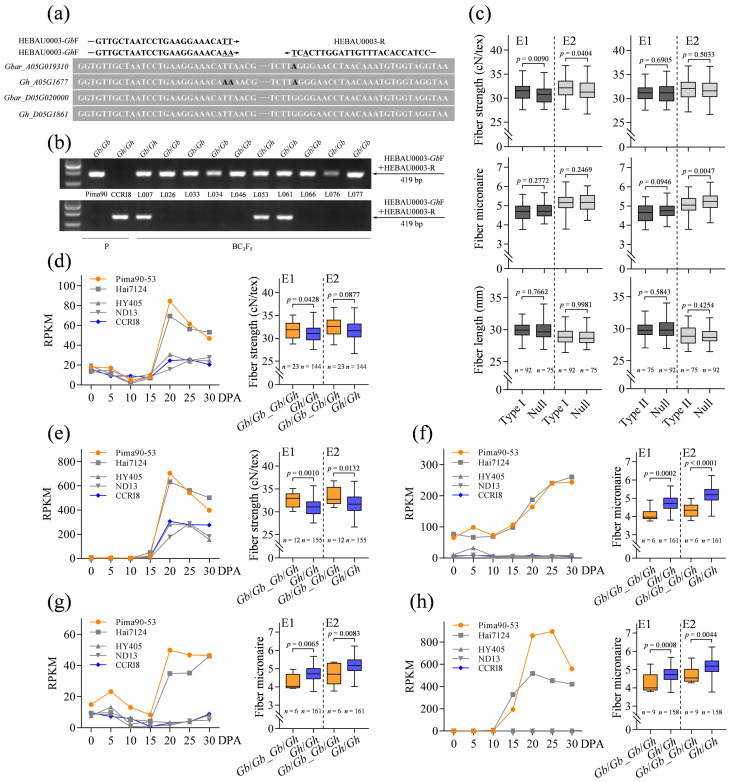
Functional analysis of the SCW biosynthesis-related candidate genes using a BC_3_F_5_ population. (**a**) Strategy for SNP marker development. (**b**) Genotyping of the BC_3_F_5_ population. (**c**) The influence on fiber properties with the introgression of candidate genes into *G. hirsutum*. In the boxplots, the center line indicates the median, box limits denote the upper and lower quartiles and whiskers mark the range of data; *n* shows the number of BC_3_F_5_ lines containing the corresponding *G. barbadense* genomic fragments. The significance of difference was analyzed with a two-tailed *t*-test. (**d**) Expression patterns of *Gbar_A05G019310* and its *G. hirsutum* ortholog and the influence on fiber strength generated by the introgression of *Gbar_A05G019310*. (**e**) Expression patterns of *Gbar_A05G025000* and its *G. hirsutum* ortholog and the influence on fiber strength generated by the introgression of *Gbar_A05G025000*. (**f**) Expression patterns of *Gbar_A06G015550* and its *G. hirsutum* ortholog and the influence on fiber micronaire generated by the introgression of *Gbar_A06G015550*. (**g**) Expression patterns of *Gbar_A13G023450* and its *G. hirsutum* ortholog and the influence on fiber micronaire generated by the introgression of *Gbar_A13G023450*. (**h**) Expression patterns of *Gbar_D05G026930* and its *G. hirsutum* ortholog and the influence on fiber micronaire generated by the introgression of *Gbar_D05G026930*.

**Table 1 ijms-24-08293-t001:** Elongation-related candidate genes.

Type	Reference ^a^	Orthologous Genes ^b^	Functional Annotation
IV-I	Gh_A03G0826	Gbar_A03G010660	GPI-anchored adhesin-like protein
	Gh_A07G1193	Gbar_A07G013810	S-adenosylmethionine synthase
	Gh_A11G1313	Gbar_A11G014750	Stomagen
	Gh_A11G3137	Gbar_A11G021290	3-hydroxyacyl-CoA dehydratase
	Gh_A13G0932	Gbar_A13G012640	SMALLER WITH VARIABLE BRANCHES
	Gh_D03G0078	Gbar_D03G000870	*α*-mannosidase
	Gh_D05G3507	Gbar_D05G037750	(+)-*δ*-cadinene synthase
	Gh_D08G1067	Gbar_D08G011240	BAHD acyltransferase
	Gh_D09G0438	Gbar_D09G004860	Subtilisin-like protease
	Gh_D11G1462	Gbar_D11G015570	Stomagen
IV-II	Gh_A03G2157	Gbar_A03G019080	Nonspecific lipid-transfer protein
	Gh_A05G1646	Gbar_A05G018970	*β*-mannanase
	Gh_A05G1812	Gbar_A05G020690	TWISTED SEED
	Gh_A07G0152	Gbar_A07G001880	Pectinesterase/pectinesterase inhibitor
	Gh_A07G0664	Gbar_A07G007890	*α*-expansin
	Gh_A08G0668	Gbar_A08G007900	BTB/POZ domain-containing protein
	Gh_A09G1368	Gbar_A09G015680	S-adenosylmethionine synthase
	Gh_A11G0547	Gbar_A11G006130	Serine protease inhibitor
	Gh_A12G1104	Gbar_A12G013240	Ribonuclease H-like protein
	Gh_D04G0006	Gbar_D04G000050	Pectinesterase/pectinesterase inhibitor
	Gh_D05G2968	Gbar_D05G031740	Importin-*β*
	Gh_D07G1581	Gbar_D07G017340	Acidic leucine-rich nuclear phosphoprotein
	Gh_D08G0151	Gbar_D08G001620	Dirigent protein
	Gh_D09G1136	Gbar_D09G012750	Alpha carbonic anhydrase
	Gh_D10G1363	Gbar_D10G014320	Methylenetetrahydrofolate reductase
	Gh_D11G0625	Gbar_D11G006420	Fatty acid amide hydrolase
	Gh_D11G3426	Gbar_D11G024050	Glutathione S-transferase
	Gh_D12G2436	Gbar_D12G026630	Heat shock protein
IV-III	Gh_A02G1644	Gbar_A02G018320	*α*-mannosidase
	Gh_A05G0923	Gbar_A05G010310	Heavy metal transport
	Gh_A05G3172	Gbar_A05G037170	ANTIAUXIN-RESISTANT
	Gh_A06G1116	Gbar_A06G013440	Cystathionine *β*-synthase
	Gh_A07G2014	Gbar_A07G023310	Formate dehydrogenase
	Gh_A10G0126	Gbar_A10G001450	Actin cytoskeleton-regulatory complex
	Gh_D04G0153	Gbar_D04G001950	Cytochrome P450
	Gh_D04G1621	Gbar_D04G017890	Cytosolic cyclophilin
	Gh_D05G2178	Gbar_D05G023090	Sugar transport protein
	Gh_D08G2441	Gbar_D08G025770	Leucine-rich repeat (LRR) protein
	Gh_D09G1484	Gbar_D09G016550	Short-chain dehydrogenase
	Gh_D11G1941	Gbar_D11G020770	PLC-like phosphodiesterase
	Gh_D12G0289	Gbar_D12G003220	SAUR-like auxin-responsive protein
	Gh_D12G0652	Gbar_D12G007360	Membrane-associated kinase regulator

^a^ Gene identifiers in the *G. hirsutum* acc. TM-1 genome [29]; ^b^ gene identifiers in the *G. barbadense* acc. 3-79 genome [2].

**Table 2 ijms-24-08293-t002:** Type I SCW biosynthesis-related candidate genes.

Reference ^a^	Orthologous Genes ^b^	Functional Annotation	Genetic Marker
Gh_A04G0686	Gbar_A04G008430	Soluble inorganic pyrophosphatase	HEBAU0001
Gh_A05G0370	Gbar_A05G004360	F-box/LRR-repeat protein	HEBAU0002
Gh_A05G1677	Gbar_A05G019310	C2H2-like zinc finger protein	HEBAU0003
Gh_A05G1930	Gbar_A05G022250	EXORDIUM protein	NA
Gh_A05G2177	Gbar_A05G025000	Transmembrane protein	HEBAU0004
Gh_A06G1590	Gbar_A06G019480	Leucine-rich repeat protein kinase	HEBAU0005
Gh_A10G1937	Gbar_A10G023160	UDP-Xyl synthase	HEBAU0006
Gh_A11G2936	Gbar_A11G034900	Laccase	HEBAU0007
Gh_A12G0165	Gbar_A12G001790	Nucleobase–ascorbate transporter	HEBAU0008
Gh_A13G0090	GB_A13G0103	Cysteine/histidine-rich DC1 domain	HEBAU0009
Gh_A13G1919	Gbar_A13G023580	Fasciclin-like arabinogalactan protein	HEBAU0010
Gh_D01G1187	Gbar_D01G013340	Protein of unknown function	HEBAU0011
Gh_D02G0286	GB_D02G0284	Cytochrome P450	HEBAU0012
Gh_D02G1423	Gbar_D02G015210	Plasmodesmata-located protein	NA
Gh_D03G0537	Gbar_D03G005540	Cytochrome P450	HEBAU0013
Gh_D03G1074	Gbar_D03G011820	Feruloyl-CoA transferase	HEBAU0014
Gh_D04G1519	Gbar_D04G016960	2-nitropropane dioxygenase	HEBAU0015
Gh_D05G1062	Gbar_D05G011330	LIGHT SENSITIVE HYPOCOTYLS	HEBAU0016
Gh_D05G2936	Gbar_D05G031310	Phosphate transporter	HEBAU0017
Gh_D06G2276	Gbar_D06G000230	*β*-tubulin	HEBAU0018
Gh_D07G0285	Gbar_D07G003310	Clathrin assembly protein	NA
Gh_D09G0011	Gbar_D09G000120	Protein of unknown function	HEBAU0019
Gh_D10G1036	Gbar_D10G011020	Xylan glucuronosyltransferase	HEBAU0020
Gh_D10G1767	Gbar_D10G018450	*β*-xylosidase	HEBAU0021
Gh_D12G2377	Gbar_D12G025300	Protein of unknown function	NA

^a^ Gene identifiers in the *G. hirsutum* acc. TM-1 genome [29]; ^b^ gene identifiers in the *G. barbadense* acc. 3-79 [2] or Hai7124 genome [3]. Note: NA, not available.

**Table 3 ijms-24-08293-t003:** Type II SCW biosynthesis-related candidate genes.

Reference ^a^	Orthologous Genes ^b^	Functional Annotation	Genetic Marker
Gh_A03G1620	Gbar_A03G020290	Progesterone 5*β*-reductase	HEBAU0022
Gh_A05G0912	Gbar_A05G010210	BEL1-like homeodomain protein	HEBAU0023
Gh_A05G1599	Gbar_A05G018470	Endo-*β*-1, 4-glucanase	HEBAU0024
Gh_A06G0882	Gbar_A06G010210	Transmembrane protein	HEBAU0025
Gh_A06G1256	Gbar_A06G015550	Aldehyde dehydrogenase	HEBAU0026
Gh_A07G0322	Gbar_A07G004070	Cellulose synthase	NA
Gh_A09G0389	NA	Protein of unknown function	NA
Gh_A10G0145	Gbar_A10G001660	NTF2-like protein	NA
Gh_A10G0696	Gbar_A10G008340	ABC transporter	HEBAU0027
Gh_A12G0108	Gbar_A12G001190	Protein of unknown function	HEBAU0028
Gh_A12G1564	Gbar_A12G018330	SLH domain protein	HEBAU0029
Gh_A12G2552	Gbar_A12G010170	Hydroxyproline-rich glycoprotein	HEBAU0030
Gh_A13G0149	Gbar_A13G001740	Auxin influx transporter	HEBAU0031
Gh_A13G0378	Gbar_A13G004310	Leucine-rich repeat (LRR) protein	HEBAU0032
Gh_A13G1904	Gbar_A13G023450	Epoxide hydrolase	HEBAU0033
Gh_A13G2039	Gbar_A13G024860	Networked (NET) family protein	HEBAU0034
Gh_A13G2237	NA	Protein of unknown function	NA
Gh_D03G0611	Gbar_D03G007510	Cellulose synthase	HEBAU0035
Gh_D05G2536	Gbar_D05G026930	Blue copper protein	HEBAU0036
Gh_D06G1911	Gbar_D06G019950	RING-type E3 ubiquitin ligase	HEBAU0037
Gh_D08G1431	Gbar_D08G015060	Subtilisin-like protease	HEBAU0038
Gh_D10G1810	Gbar_D10G018890	UDP-glycosyltransferase	NA
Gh_D12G0575	Gbar_D12G006270	Kinesin-like protein	HEBAU0039

^a^ Gene identifiers in the *G. hirsutum* acc. TM-1 genome [29]; ^b^ gene identifiers in the *G. barbadense* acc. 3-79 genome [2]. Note: NA, not available.

## Data Availability

The data presented in this study are available within the article and the Appendix A. The raw RNA-seq data have been deposited in the Genome Sequence Archive (https://bigd.big.ac.cn/gsa; PRJCA014806 (accessed on 26 April 2023)) and are available from the corresponding author on reasonable request.

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
