# Peer review of "Transcriptome, Ectopic Expression and Genetic Population Analysis Identify Candidate Genes for Fiber Quality Improvement in Cotton"

_ijms, 2023, doi:10.3390/ijms24098293_

Round 1

Reviewer 1 Report

The manuscript conducted comprehensive comparisons between G. barbadense and G. hirsutum of seven fiber development stages.They identified and analyzed a batch of preferentially expressedgenes during fiber development in G. barbadense, and thus further revealed the genetic basis for high-quality cotton fiber formation. More importantly, they identified 42 elongation-related and 48 SCW biosynthesis-related candidate genes for cotton fiber quality improvement. The potential of candidate genes was then confirmed by transgene and SNP marker-based BC3F5 population analysis. The results not only provide useful insights into cotton fiber development, but also credibly provide targets for genetic manipulation and marker-assisted multigene introgression to improve fiber quality.

Generally, the experiments were well designed, and the manuscript was well prepared. The Introduction, Material & Methods and Results sections as well as the Discussion section were written well. However, this manuscript also has some flaws that require a minor revision before it can be accepted for publication.

In general, there are frequent errors with respect to English expression that need to be addressed by having the manuscript edited by a professional editorial service or a native English-speaker before re-submission.

This part of “Abstract” in the manuscript needs to be further streamlined to extract the highlights of the full text.
This part of DISCUSSION suffers from verbosity and requires editing to make it better. In addition, there are some errors with respect to English expression. I suggest that the manuscript be viewed by a native English speaker or an editorial service after revision.

Author Response

In general, there are frequent errors with respect to English expression that need to be addressed by having the manuscript edited by a professional editorial service or a native English-speaker before re-submission.

Our response: Thanks for your comment and the point was taken. The manuscript has been edited carefully to better follow English expression.

This part of “Abstract” in the manuscript needs to be further streamlined to extract the highlights of the full text.

Our response: Thanks for your suggestion and the point was taken. The “Abstract” section has been reorganized to make the highlights stand out. The text is as follows.

Abstract: Comparative transcriptome analysis of fiber tissues between Gossypium barbadense and Gossypium hirsutum could reveal molecular mechanisms underlying high-quality fiber formation and identify candidate genes for fiber quality improvement. In this study, 759 genes were found to be firmly up-regulated at the elongation stage in G. barbadense, which showed four distinct expression patterns (I-IV). Among them, the 346 genes of group IV stood out for the potential in promoting fiber elongation, in which we finally identified 42 elongation-related candidate genes by comparative transcriptome analysis between G. barbadense and G. hirsutum. Subsequently, we overexpressed GbAAR3 and GbTWS1, two of the 42 candidate genes, in Arabidopsis plants and validated their roles in promoting cell elongation. At the secondary cell wall (SCW) biosynthesis stage, 2275 genes were consistently up-regulated and exhibited five different expression profiles (I-V) in G. barbadense. We highlighted the critical roles of the 647 genes of group IV in SCW biosynthesis and further picked out 48 SCW biosynthesis-related candidate genes by comparative transcriptome analysis. SNP molecular markers were then successfully developed to distinguish the SCW biosynthesis-related candidate genes from their G. hirsutum orthologs, and genotyping and phenotyping of a BC3F5 population proved their potential in improving fiber strength and micronaire. Our results contribute to better understanding of the fiber quality differences between G. barbadense and G. hirsutum and provide novel alternative genes for fiber quality improvement.

This part of DISCUSSION suffers from verbosity and requires editing to make it better. In addition, there are some errors with respect to English expression. I suggest that the manuscript be viewed by a native English speaker or an editorial service after revision.

Our response: Thanks for your comment and the point was taken. The manuscript has been reviewed carefully and a number of grammatical errors have been revised.

Reviewer 2 Report

The article entitled "Transcriptome, ectopic expression and genetic population analysis identify candidate genes for fiber quality improvement in cotton" is well compiled manuscript, and the authors combined the RNA-seq data from different allotetraploid Gossypium species and genetic transform technology on Arabidopis thaliana and high-generation segregation population to screen and confirm the candidate genes that could affect fiber development. In general, the results are innovative, significant and useful for the cotton research. However, several technical issues should be addressed first. 

(1) The cited references in this manuscript might be incorrect, since other articles utilized the Arabic numbers. Besides, we noticed there were no the latest references, especially in 2022 and 2023.

(2) The total of 28 RNA samples were sequenced in this study, while there was no biological repeats. How could the authors guarantee the accuracy of the RNA-seq data ?

(3) Hu et al (2019), Wang et al (2019), and Huang et al (2020) published the third-generation genome data, Why the authors chose the 2015 type ? Suggest to utilized the 2019-type cotton genome by Hu et al to mapping the results, since they sequenced the TM-1 and Hai7124.

Author Response:

The article entitled "Transcriptome, ectopic expression and genetic population analysis identify candidate genes for fiber quality improvement in cotton" is well compiled manuscript, and the authors combined the RNA-seq data from different allotetraploid Gossypium species and genetic transform technology on Arabidopis thaliana and high-generation segregation population to screen and confirm the candidate genes that could affect fiber development. In general, the results are innovative, significant and useful for the cotton research. However, several technical issues should be addressed first. 

(1) The cited references in this manuscript might be incorrect, since other articles utilized the Arabic numbers. Besides, we noticed there were no the latest references, especially in 2022 and 2023.

Our response: Thanks for your comment and the point was taken. We have revised the "References" section and utilized the Arabic numbers to indicate the citation.

(2) The total of 28 RNA samples were sequenced in this study, while there was no biological repeats. How could the authors guarantee the accuracy of the RNA-seq data ?

Our response: Thanks for your comment. We have also noticed this deficiency and therefore identified DEGs (differentially expressed genes) whose expression was consistent in two G. barbadense cultivars as well as in two G. hirsutum cultivars. In fact, we have demonstrated the accuracy of the RNA-seq data in a previous article entitled “Genome-wide identification and analysis of GH9 gene family in Gossypium barbadense L.” (DOI: 10.13304/j.nykjdb.2021.0312), which presented the GH9 gene expression based on both the RNA-seq data and the qRT-PCR data (Fig. A1). Additionally, ectopic expression and genetic population analysis have been employed to strengthen the confidence of the fiber development-related candidate genes.

Fig. A1 Expression profiles of GbGH9s during cotton fiber development in Hai7124. The qRT-PCR data represent the total expression levels of the two highly homologous genes from At- and Dt-subgenome.

(3) Hu et al (2019), Wang et al (2019), and Huang et al (2020) published the third-generation genome data, Why the authors chose the 2015 type ? Suggest to utilized the 2019-type cotton genome by Hu et al to mapping the results, since they sequenced the TM-1 and Hai7124.

Our response: Thank you for pointing this out. It will be a long process to map the reads and then achieve the manuscript, for we need to reanalyze the generated data and reorganize our figures and tables. As a replacement, we have identified the orthologous genes in G. barbadense acc. 3-79 (Wang et al., 2019) or Hai7124 (Hu et al., 2019) for the 42 elongation-related candidate genes (Table 1) and 48 SCW biosynthesis-related candidate genes (Table 2 and Table 3), as well as for the 759 up-regulated genes at the elongation stage (Table S3) and 2275 up-regulated genes at SCW biosynthesis stage (Table S5). The corresponding references have also been cited.

Reviewer 3 Report

Dear Authors,

Your work is well reflected by the results in the manuscripts and I think you did a good job by approaching a topic of interest. However, to improve the quality, I suggest the following:

Line 515: I suggest the exact mention of the period. From those presented in Material and Method, the year, years of study are missing. The location in real time of the research is not clear, so what you should write in brackets is the year/s in which you made the observations.

Line 523: It would be appreciated if you could briefly mention what area you used to grow the plants of the 2 species of cotton.

Line 525: Explain what local farming methods you are referring to. It is not clear, from what is reported in the paper (in fact, the culture part is presented vaguely), in which system the plants were cultivated, considering their diversity. It is necessary for readers all over the world who do not know the system applied in your region. Here I mean the classic system, organic, irrigated, non-irrigated, covered, uncovered (in open field)...or others.

Line 612: Something is not very clear. The conclusions refer only to G. barbadense, while in your studies you focus on comparisons between G. barbadense and G. hirsutum of seven 19 fiber development stages. There should be a connection between the abstract and the conclusions, that's why I suggest you reformulate the conclusions so that they reflect the essence of the studies.

Line 625: Please write the full name of the journal to be in line with the other references.

Line 736: What is the abbreviation of G3? Make sure reference 40 is spelled correctly. After searching for this reference, I did not find it online. Make sure it is accessible, if it exists.

All these suggestions can be found directly in the attached manuscript.

Kind regards, R

Author Response

Line 515: I suggest the exact mention of the period. From those presented in Material and Method, the year, years of study are missing. The location in real time of the research is not clear, so what you should write in brackets is the year/s in which you made the observations.

Our response: Thanks for your comment and the point was taken. In fact, we have reorganized the “Plant materials and growth conditions” subsection and the text is as follows.

4.1 Plant materials and growth conditions

Cotton plant materials used in this study included G. barbadense acc. Pima90-53 [77] and Hai7124 [3], and G. hirsutum acc. HY405 [35], ND13 [35], and CCRI8. Pima90-53 (accession number: M210080; introduced from USA), Hai7124 (M210054; Jiangsu, China), and CCRI8 (M110553; Henan, China) were collected and preserved, with the appropriate permissions, by the National Medium-term Gene Bank of Cotton in China. HY405 (G100937; Hebei, China) and ND13 (G100728; Hebei, China) were collected and preserved by Hebei Agricultural University. All necessary permissions for planting and investigating these cultivars were obtained from Hebei Agricultural University and the National Medium-term Gene Bank of Cotton in China. The collection and research of these cultivars have complied with the Convention on the Trade in Endangered Species of Wild Fauna and Flora. A BC3F5 population consisting of 167 lines was constructed and used in this study. Firstly, G. barbadense acc. Pima90-53, as the male parent, was crossed with G. hirsutum acc. CCRI8 to generate F1 hybrid in Baoding, Hebei Province, China, during summer 2009. The hybrid plants, as the female parent, were then continuously backcrossed with CCRI8 for three generations to produce BC3F1 plants in Baoding from 2010 to 2012. Eventually, to raise BC3F5 plants, the BC3F1 plants were self-pollinated for four generations in summer 2013 (Baoding), winter 2013 (Sanya, Hainan, China), summer 2014 (Baoding), and summer 2015 (Baoding). Molecular marker-assisted selection was performed at BC3F1-BC3F3 generations to ensure the introgression of G. barbadense into G. hirsutum.

For transcriptome sequencing, Pima90-53, Hai7124, HY405, and ND13 were grown at the cotton breeding center (38°45′N, 115°29′E) in Baoding, with a warm temperate continental monsoon climate, from late April to late October in 2014. Each plot consisted of six rows of 7 m in length containing 20-22 plants per row, with 30-35 cm between plants within each row and 80 cm between rows. The 167 BC3F5 lines and their parents were planted in a randomized complete block design including three replications in Qingxian, Hebei Province (38°65′N, 116°91′E; with a temperate semi humid continental monsoon climate), and in Luntai, Xinjiang Uygur Autonomous Region (41°46′N, 84°14′E; with a warm temperate continental arid climate) in 2016. In Qingxian, each plot contained one row of 7 m in length, with 20-22 plants per row, 30-35 cm between plants within each row, and 76 cm between rows, whereas in Luntai each plot contained one row of 7 m in length containing 60-66 plants per row, with about 10 cm between plants within each row and 76 cm between rows. The population materials were sown in mid to late April and were harvested in mid to late October. All field management including watering, weed management, and fertilization was performed according to local production practices throughout the growth period. A. thaliana (Columbia ecotype) wild-type and transgenic plants were grown in pots containing sterile vermiculite in a greenhouse (22°C, 16-h photoperiod, and 70% relative humidity) in Hebei Agricultural University, with Hoagland’s nutrient solution added weekly.

Line 523: It would be appreciated if you could briefly mention what area you used to grow the plants of the 2 species of cotton.

Our response: Thanks for your comment and the point was taken. For details, please refer to our responses to the Line 515.

Line 525: Explain what local farming methods you are referring to. It is not clear, from what is reported in the paper (in fact, the culture part is presented vaguely), in which system the plants were cultivated, considering their diversity. It is necessary for readers all over the world who do not know the system applied in your region. Here I mean the classic system, organic, irrigated, non-irrigated, covered, uncovered (in open field)...or others.

Our response: Thanks for your comment and the point was taken. For details, please refer to our responses to the Line 515.

Line 612: Something is not very clear. The conclusions refer only to G. barbadense, while in your studies you focus on comparisons between G. barbadense and G. hirsutum of seven 19 fiber development stages. There should be a connection between the abstract and the conclusions, that's why I suggest you reformulate the conclusions so that they reflect the essence of the studies.

Our response: Thanks for your suggestion and the point was taken. We have reformulated the "Conclusions" section and the text is as follows.

In the present study, we highlighted a batch of genes preferentially expressed during fiber elongation and SCW biosynthesis, and thus further revealed the genetic basis underlying high-quality fiber formation. By comparative transcriptome analysis between G. barbadense and G. hirsutum, we finally identified 42 elongation-related and 48 SCW biosynthesis-related candidate genes, whose potential roles were then confirmed by ectopic overexpression and SNP marker-based BC3F5 population analysis, respectively. These findings not only provide valuable information for the understanding of cotton fiber development, but also credibly contribute to cotton breeding for fiber quality improvement.

Line 625: Please write the full name of the journal to be in line with the other references.

Our response: Thanks for your comment and the point was taken. The "References" section has been reorganized to conform to the formats of IJMS, and in this process several errors have been revised.

Line 736: What is the abbreviation of G3? Make sure reference 40 is spelled correctly. After searching for this reference, I did not find it online. Make sure it is accessible, if it exists.

Our response: Thanks for your comment. The full name of G3 is “G3-Genes Genomes Genetics” and we have revised this citation. The link of full text is as follows.

https://doi.org/10.1534/g3.117.300108